# Steering Your Generalists: Improving Robotic Foundation Models via Value Guidance

**Mitsuhiko Nakamoto**[1]    **Oier Mees**[1]    **Aviral Kumar**[2,3]    **Sergey Levine**[1]

[1]UC Berkeley    [2]Carnegie Mellon University    [3]Google DeepMind

https://nakamotoo.github.io/V-GPS

**Abstract:** Large, general-purpose robotic policies trained on diverse demonstration datasets have been shown to be remarkably effective both for controlling a variety of robots in a range of different scenes, and for acquiring broad repertoires of manipulation skills. However, the data that such policies are trained on is generally of mixed quality – not only are human-collected demonstrations unlikely to perform the task perfectly, but the larger the dataset is, the harder it is to curate only the highest quality examples. It also remains unclear how optimal data from one embodiment is for training on another embodiment. In this paper, we present a general and broadly applicable approach that enhances the performance of such generalist robot policies at deployment time by re-ranking their actions according to a value function learned via offline RL. This approach, which we call Value-Guided Policy Steering (V-GPS), is compatible with a wide range of different generalist policies, without needing to fine-tune or even access the weights of the policy. We show that the same value function can improve the performance of five different state-of-the-art policies with different architectures, even though they were trained on distinct datasets, attaining consistent performance improvement on multiple robotic platforms across a total of 12 tasks.

**Keywords:** Generalist Policies, Value Functions, Robot Reinforcement Learning

## Test-time Policy Steering via Value Guidance

Figure 1: **(V-GPS)** We introduce Value-Guided Policy Steering (V-GPS), a novel approach that improves the performance of pre-trained generalist robotic policies by re-ranking their actions at deployment time based on a value function learned via offline RL. The same single V-GPS value function can be combined with any off-the-shelf generalist policy in a plug-and-play manner, without the need to fine-tune or access the policy's weights, improving downstream performance across multiple robotic platforms.

8th Conference on Robot Learning (CoRL 2024), Munich, Germany.

# 1 Introduction

Large, high-capacity models trained on diverse datasets are key to the effectiveness of modern machine learning methods [1, 2, 3, 4, 5]. However, this recipe presents a major challenge in robotic learning: while large datasets collected from diverse sources have recently made the study of large-scale robotic learning feasible [6, 7], such data sources are typically of mixed quality, and recovering high-performing and fluent policies from such suboptimal data presents a major challenge. While state-of-the-art imitation learning methods can effectively replicate the distribution of demonstrations [8, 9, 6], the mixed quality of these datasets makes this distribution fall short of the performance we would like from our robotic systems. More concretely, generalist policies often fail due to imprecise manipulation, such as failed grasping or early dropping, despite their strong semantic generalization (as is evident from a number of existing results; see Appendix G of Brohan et al. [8] & Section 5.2 of Black et al. [10]). These issues become more severe when the policy encounters environmental distribution shifts, even if these shifts are extraneous in nature (e.g., changes in table texture, camera pose, and distractors; see Figures 8 and 9 of Li et al. [11]).

How can we improve the precision, robustness, and proficiency of these generalist policies while still retaining the best benefits of their scale and generalization capabilities? While one intuitive method could involve refining the policy through fine-tuning, this approach is infeasible in the non-stationary real world. Not only would each adaptation cycle require costly human tele-operated or instrumented data collection, but also unintuitive hyperparameter tuning to prevent the model from losing its generalist capabilities. If we can instead devise an approach to preserve the generalist policy, but simply "steer" it in a way that improves precision and robustness upon deployment, that would be most desirable. What is a good way to steer an off-the-shelf generalist policy? Our key insight is that re-ranking multiple action proposals from a generalist policy using a value function at test-time allows us to accomplish this. Re-ranking with some sort of value or reward functions is extremely effective at improving the reasoning capabilities of large language models (LLMs) [12, 13, 14] in single-step "bandit" problems. However, it is yet to be shown effective for multi-step robotic manipulation problems with stochastic environment interaction from raw pixel observations, outside of simulated gym environments [15, 16].

In this paper, we build a recipe to train a general robotic value function via offline reinforcement learning (RL) and show that despite the multi-step nature of robotic problems, value-function guided test-time action selection is an effective approach for improving generalist policies. As shown in Figure 1, this approach enables us to directly improve the generalist policy on scenes and manipulation problems encountered at *deployment* time, unlike prior offline RL methods that use a value function on the training data and hence, still fail on shifts encountered upon deployment. In addition, using the value function at only test-time is modular and plug-and-play with any generalist policy, and does not require tuning bells and whistles as conventional robotic offline RL pipelines [17, 18].

The main contribution of this paper is V-GPS, **V**alue-**G**uided **P**olicy **S**teering, a method that uses offline RL value functions to steer generalist robotic policies. We build a recipe to pre-train a value function on diverse robotic datasets, demonstrating that their use improves maneuvers of several open-source generalist policies. To our knowledge, this is the first work to leverage test-time action sampling for real-world robotic generalist policies. We conduct extensive evaluations in both simulated [11] and real-world environments [19], using two different embodiments, across a total of 12 tasks, and on top of five state-of-the-art open-source generalist policies including Octo [9], RT1-X [6], and OpenVLA [20]. V-GPS achieves an improvement of **+82%** in real-world manipulation tasks, and consistently enhances all five generalist policies across multiple embodiments.

# 2 Related Work

**Large-scale Robotic Datasets and Policies.** Many prior works have collected and open-source robotic datasets [7, 21, 19, 22, 23, 24, 25, 26, 27, 28, 29]. These datasets include various quality of data that have been collected in diverse ways, ranging from human tele-operations [7, 21, 19, 26] to autonomous collection with scripted or random policies [22, 23]. Recent efforts to aggregate

these existing datasets [6] have made learning from large-scale, multi-source datasets more feasible for the community. Thus, a number of works have leveraged these large-scale datasets to train general-purpose robotic policies, which have shown generalization to controlling multiple robot manipulators with a single policy [9, 6], to unseen tasks [8, 30], and to new language instructions or goals [31, 30, 9, 32, 33, 34]. RT-X [6] and Octo [9] have built and open-sourced robotic foundation models with high generalization capability by scaling the policy with Transformer architecture [35] and training with advanced imitation learning methods [36, 37]. However, these generalist policies often fail due to imprecise manipulation, especially when the policy encounters environmental distribution shifts [10, 11]. Our work is broadly applicable to these off-the-shelf policies, aiming to improve their performance by seamlessly integrating a value function at test time.

**Value-based Offline RL for Robotics.** Prior works have suggested that offline RL can, in principle, recover more optimal behavior than imitation learning from mixed-quality data [38, 39]. For real-world robotic tasks, several studies have also shown offline RL to be effective for scaling with large datasets [17, 40, 19] and large models [18, 41]. RL policies trained with value functions [42] can naturally learn to predict actions that maximize long-term rewards [43]. However, these methods typically leverage standard model-free offline RL algorithms [44, 45, 46, 47, 48, 49] that require using value functions to train and update the policy, so that the policy models are often limited to Gaussian distributions. This makes it difficult to scale the policy to state-of-the-art expressive architectures or to leverage pre-trained generalist robotic policies. In contrast, our method provides a more general and flexible way to leverage the value function that can be integrated with any off-the-shelf, black-box pre-trained policy.

**Sampling-based Action Selection.** Another way to leverage value functions is to use them for sampling-based action selection. In this approach, multiple actions are sampled from the policy and then ranked using the value function, with the top actions selected and executed. For language models, sampling-based action selection has been shown to be effective in improving performance for tasks such as Q&A and summarization [12, 13, 14, 50, 51]. In the robotics domain, Brohan et al. [52] demonstrates the effectiveness of using a language-grounded value function and employs it to score high-level language commands in skill space. However, their focus is to enhance high-level reasoning rather than improve low-level performance. Prior work [15, 53, 16] has also shown that training a value function to directly score low-level actions is effective on D4RL simulation tasks [54]. However, this approach has not yet been applied to diverse, high-dimensional, real-world robotic tasks. Our work is the first to show that training value functions on real-world robotic datasets can effectively guide sampling-based low-level action selection, leading to improvements in large-scale robotic foundation models.

## 3 Preliminaries and Problem Statement

We study problems where we wish to control a robot through language instructions. We assume access to a generalist, language-conditioned robotic policy $\pi(a \mid s_t, l)$, which can sample multiple actions $a_1, \ldots, a_K$ given the current state $s_t$ and a language command $l$. Note that we do not assume access to the model weights of $\pi$, allowing the policy to be completely black-box.

For building our approach, we will consider the formalism of a Markov decision process (MDP) $\mathcal{M} = (\mathcal{S}, \mathcal{A}, P, r, \gamma)$. $\mathcal{S}, \mathcal{A}$ denote the state and action spaces, and $P(s'|s, a)$ and $r(s, a)$ denote the dynamics and reward functions. $\gamma \in (0, 1)$ denotes the discount factor. The value function $Q(s, a)$ represents the long-term discounted return $\sum_t \gamma^t R(s_t, a_t)$. Our approach will prescribe a recipe to learn this value function and then show that it is helpful in steering a pre-trained generalist policy. In our setting, the dataset is a language-annotated robot dataset $\mathcal{D} = \{(\tau^1, l^1), (\tau^2, l^2), \ldots, (\tau^N, l^N)\}$, where each trajectory $\tau^n$ consists of a sequence of states $\mathbf{s}_i^n \in \mathcal{S}$ and actions $\mathbf{a}_i^n \in \mathcal{A}$ along with a natural language command $l^n$ describing the task performed in the trajectory.

## 4 Analysis: Failure Modes of Generalist Policies

To motivate the failure modes of generalist policies and develop our approach, we begin by investigating failure modes associated with generalist robotic manipulation policies. For this analysis, we use the Octo-small-1.5 model [9], an open-source transformer-based generalist robotic policy

trained on the OXE dataset and attempt to investigate some failure modes of this policy. The videos can be found at https://nakamotoo.github.io/V-GPS.

**Case 1: Failure of precise grasping.** We first evaluate the Octo policy on a real WidowX robot platform for the task "put pepper in pot" (see Scene A in Figure 3). The surface of the plastic green pepper is slippery and presents an uneven curvature, often making it critical to choose the grasp point and magnitude of the gripper action appropriately for a reliable grasp (see Figure 2). Even when policies can grasp the green pepper, imperfect grasp points or gripper actions often lead to the object falling off the gripper while the task is being executed.

**Case 2: Pre-maturely timed attempts to complete the task.** We conduct an additional study on the task "put mushroom on cloth," as shown in Figure 2. Unlike the green pepper, the mushroom is relatively easier to grasp because it's a soft object. In our evaluation, we found that the Octo policy is indeed able to successfully grasp the object and move it towards the cloth. However, it tends to drop the mushroom pre-maturely, such that the mushroom does not land on the cloth. In addition to such premature attempts to complete the task, we also observe cases where Octo does not release the object in a timely

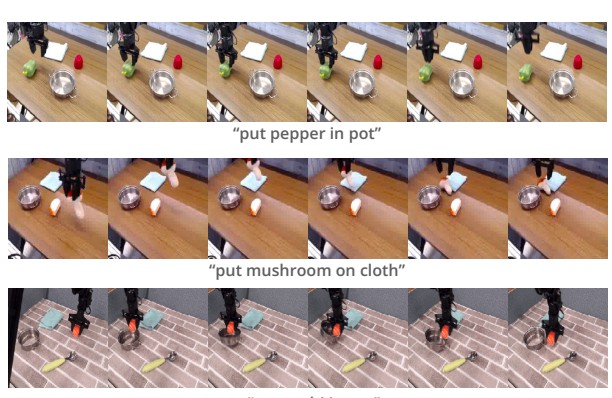

"put pepper in pot"

"put mushroom on cloth"

"put sushi in pot"

**Figure 2: (Failures of Octo)** Octo policy encounters failures such as imprecise grasping (first row), dropping the object prematurely (second row), and holding onto the object for too long (third row).

manner. Often, the target obtains and remains stuck in the gripper, and arbitrary arm drifts during this period eventually cause the object to fall outside the target container. For example, in the task "put sushi in pot" in Scene C (see Figure 3), the model tends to hold onto the sushi for too long, resulting in it being dropped outside of the container, as shown in Figure 2.

## 5 V-GPS: Value-Guided Policy Steering

While these failure cases may vary among different policies and scenarios, they highlight the room for improvement in the precision and robustness of generalist robotic policies. In this section, we will describe our approach V-GPS which utilizes a value function to improve generalist policies to avoid these failures. The key insight behind V-GPS is to use a value function to "re-rank" multiple actions sampled from the generalist pre-trained policy and execute the action that the value function thinks is most likely to succeed. We achieve this by first training a language-conditioned value function on a robotic dataset and then combining it with generalist policies at test time. We describe each of the phases of V-GPS below.

### 5.1 Training: Learning Value Function via Offline RL Pre-training

The main component of V-GPS is a language-conditioned Q function $Q_\theta(s, a, l)$, where $s$ is the state, $a$ is the action, and $l$ is the language instruction. To train such a value function, we first need to obtain a reward function that can be used to supervise the value function training. Recall that in our problem setting, we are only provided with a dataset $D$ of language-conditioned robotic data, where each entry in $D$ consists of a trajectory $\tau^i$ and a language instruction $l^i$. To convert this dataset into a form that is amenable to training a Q-function, we annotate the last $H$ transitions of this trajectory with a sparse binary reward value of $+1$ to indicate completion of the task specified by the language instruction. The reward values for all other transitions in this trajectory are marked as $0$. In our experiments, we set the value of $H = 3$ following Kumar et al. [17], which utilized an analogous scheme for learning value functions but with one-hot task descriptors.

Equipped with this reward function, in principle, one can use any offline RL or policy evaluation algorithm to fit $Q_\theta$. Each algorithm will fit the value function of a different policy: while algorithms

such as SARSA [55] will fit the value function of the behavior policy, "full" offline RL methods such as conservative Q-learning (CQL) [46], calibrated Q-learning (Cal-QL) [56], and implicit Q-learning (IQL) [47] attempt to find the optimal value function supported on actions sampled from the behavior policy. Our intended approach, which uses value functions for re-ranking, presents some unique requirements: **(1)** since we only utilize the value function *once* to re-rank actions from the generalist policy (as opposed to iterative re-ranking), we need this value function to be as close as possible to the optimal value function; and **(2)** since we wish to use one single value function for steering multiple generalist policies that are trained on large-scale diverse datasets, we want our value function to be robust to out-of-distribution (OOD) actions. Given these requirements, we choose to utilize Cal-QL [56] as our main algorithm for training the value function, as it attempts to approximate the "best in support" approximation of the optimal value function while being robust to noisy actions due to its conservative objective. While we use Cal-QL for our main algorithm, we demonstrate that IQL is also effective for V-GPS in Appendix A.

Formally, Cal-QL trains a Q function $Q_\theta(s, a, l)$ with the following objectives, where $\mathcal{B}^\pi Q_{\bar\theta}$ is the backup operator applied to a delayed target Q-network $Q_{\bar\theta}$, $Q^\mu(s, a, l)$ is the Q-value of a reference policy $\mu$, and $\alpha$ is a hyperparameter to control the conservative penalty. The conservative regularizer aims to penalize the learned Q-function for OOD actions while compensating for this pessimism on actions seen in the training dataset.

$$J_Q(\theta) = \alpha \underbrace{\left(\mathbb{E}_{s,l\sim\mathcal{D},a\sim\pi}\left[\max\left(Q_\theta(s,a,l), Q^\mu(s,a,l)\right)\right] - \mathbb{E}_{s,a,l\sim\mathcal{D}}\left[Q_\theta(s,a,l)\right]\right)}_{\text{Calibrated conservative regularizer } \mathcal{R}(\theta)}$$
$$+ \frac{1}{2}\mathbb{E}_{s,a,s',l\sim\mathcal{D}}\left[\left(Q_\theta(s,a,l) - \mathcal{B}^\pi Q_{\bar\theta}(s,a,l)\right)^2\right]. \tag{1}$$

**Implementation details.** In our experiments, we had to utilize several design choices and implementation details to obtain a good value function. While a complete set of hyperparameters is provided in the Appendix B, here we discuss some central design choices.

**(i) Reward function.** As discussed above, since our offline datasets do not specify reward annotations, we label the last $H$ steps of a demonstration rollout with a +1 reward. Following the binary reward scheme from Kumar et al. [17], where we choose $H = 3$. In addition, instead of utilizing reward values 0 and 1, we found it better to utilize shifted reward values of 0 and $-1$.

**(ii) Model architecture.** Our language-conditioned value function $Q_\theta(s, a, l)$ uses a ResNet-34 [57] image encoder with FiLM language conditioning. While this architecture has been shown to be effective for learning language-conditioned behavior cloning policies [19, 58, 59], we find it also effective for learning value functions. The language instructions are first processed by a frozen MUSE encoder [60], and then passed into every block in ResNet with FiLM conditioning [61].

### 5.2 Deployment: Test-Time Action Re-Ranking

Once we obtain a value function, we can use it to steer any generalist policy $\pi$ upon deployment. A simple idea for doing this would be to use the value function for "re-ranking" multiple action candidates sampled from the generalist policy. Specifically, at any given moment during deployment, given the current observation $s_t$ and the language prompt $l$, we first sample $K$ actions $\{a_1, \ldots, a_K\}$ from the generalist policy $\pi$, and query the value function $Q_\theta$ to get scores for each action candidate. Given these scores, one can choose which actions to select by either acting greedily as $a_t = \arg\max_{a_i, i=1\ldots K} Q(s, a_i)$, or sample the action from a "re-ranked" categorical distribution obtained by computing a temperature-controlled softmax over Q-values:

$$a_t \sim \text{Softmax}\left(\frac{Q_\theta(s_t, a_1)}{\beta}, \ldots, \frac{Q_\theta(s_t, a_K)}{\beta}\right), \tag{2}$$

where $\beta$ is a temperature parameter that controls the sharpness of the distribution, making the sampling process more and more greedy as $\beta \to 0$. This hyperparameter makes our method more flexible, as it allows us to strike a balance between how much we trust the policy and how much we

**Algorithm 1** V-GPS: Test-Time Action Selection & Execution

---

**Require:** Language-conditioned policy $\pi(\mathbf{a} \mid \mathbf{s}_t, \mathbf{l})$, Q-function $Q_\theta(\mathbf{s}_t, \mathbf{a}, \mathbf{l})$, initial state $\mathbf{s}_0^{\text{test}}$, language command $l^{\text{test}}$, maximum time step $T$, number of actions to sample $K$, temperature $\beta$

1: $t \leftarrow 0$
2: **while** $t \leq T$ **do**
3:      Sample $\{a_1, \ldots, a_K\} \sim \pi(\mathbf{a} \mid \mathbf{s}_t^{\text{test}}, \mathbf{l})$           ▷ Propose $K$ actions from policy
4:      Select $\mathbf{a}_t \sim \text{Softmax}\left(\frac{Q_\theta(s_t, a_1)}{\beta}, \ldots, \frac{Q_\theta(s_t, a_K)}{\beta}\right)$    ▷ Re-rank and select high-value action
5:      Execute $\mathbf{a}_t$
6:      $\mathbf{s}_{t+1}^{\text{test}} \leftarrow$ New observation
7:      $t \leftarrow t + 1$
8: **end while**

---

rely on the value function. Further details of our design choices and implementations can be found in the Appendix B. Pseudocode for test-time action re-ranking and control is provided in Algorithm 1.

## 6 Experimental Evaluation

The goal of our experiments is to evaluate the effectiveness of V-GPS in improving the robustness and precision of a number of generalist policies for open-world language-guided robotic manipulation problems. To this end, we aim to answer the following questions:

1. Can V-GPS improve the downstream performance of a number of off-the-shelf generalist policies across different embodiments?
2. What kind of failures of generalist policies does V-GPS address?

To answer these questions, we conduct evaluations in both simulated and real-world environments, using two different embodiments, across a total of 12 tasks, and on top of five state-of-the-art open-source generalist policies. Note that we use the same *single* value function trained on cross-embodiment data for all policies across both real-world and simulated tasks.

### 6.1 Experimental Scenarios and Comparisons

**Training dataset:** To apply V-GPS on cross-embodiment tasks, we trained a single value function on a mix of Bridge V2 dataset [19] and Fractal dataset [24, 6]. The Bridge V2 dataset consists of 45K language-annotated manipulation demonstrations collected in 24 environments at 5Hz. The Fractal dataset is a collection of open-world manipulation demonstrations, comprising 130K episodes that cover more than 700 tasks collected on the Google Robot.

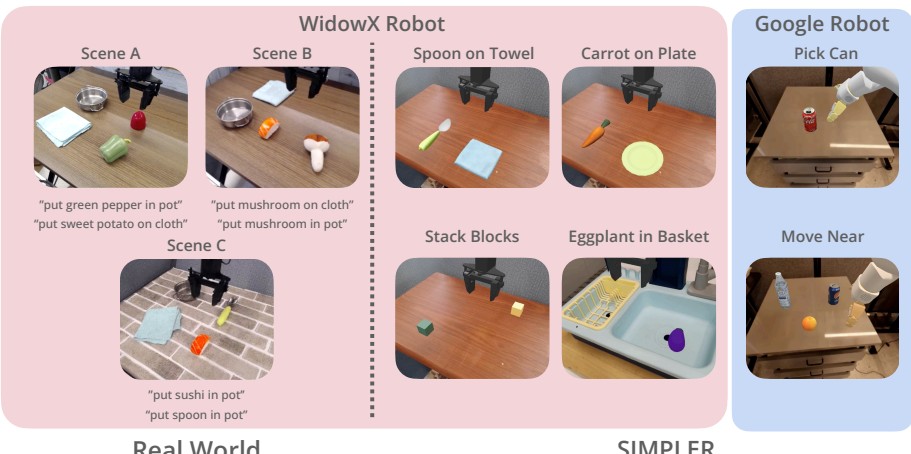

**Figure 3: (Experimental setup)** We evaluate our method on 12 tasks in total. In the real-world WidowX robot platform, we study 6 tasks across 3 different scenes. In the SIMPLER simulated evaluation suite, we study 4 tasks on the WidowX platform and 2 tasks on the Google Robot.

**Real-world setup and tasks:** We conduct our real-robot evaluations on a 6 DOF WidowX250 robot arm. Our evaluations were carried out across 6 tasks in 3 scenes as shown in Figure 3.

**Simulation setup and tasks:** Our simulated experiments are performed in the SIMPLER environment [11]. SIMPLER is a real-to-sim evaluation suite designed specifically for real-robot policies, as it can accurately reflect real-world performance. We evaluate 6 tasks on two robot embodiments: 4 tasks on the WidowX arm and 2 tasks on the Google Robot, as shown in Figure 3.

**Generalist policies:** We evaluate V-GPS on top of five different general-purpose robotic policies:

- **Octo-small** [9], a 27M parameter open-source generalist policy pre-trained on a mix of 25 different datasets from the Open-X-Embodiment (OXE) dataset [6]. The policy uses a transformer backbone based on ViT-S [62], followed by a diffusion action head to model expressive action distributions. While the model can take either a language instruction or an image as a goal, we use its language-conditioned feature for our experiments.

- **Octo-base** [9], the larger version of Octo, with 93M parameter based on ViT-B [62] backbone.

- **Octo-small-1.5** [9], the updated version of the Octo-small model. The same architecture is trained with augmented language instruction via rephrasing from GPT-3.5 and repeating the language tokens at every context window, aiming for improved language understanding.

- **RT1-X** [6], a 35M parameter transformer policy pre-trained on the OXE dataset.

- **OpenVLA** [20], a 7B parameter vision-language-action model, trained on 970k episodes of robotic demonstration from OXE dataset [6]. The policy was fine-tuned on a pre-trained vision language model, Prismatic [63].

We provide further details about the baselines and the evaluation setup in Appendix C and D.

## 6.2 What Kind of Failure Modes of the Generalist Policy Does V-GPS Address?

**Real-world results.** We present the performance on real-world tasks in Table 1. V-GPS consistently improves Octo-small-1.5 in all 6 tasks, with notable improvements of **+55%** in Scene A, **+92%** in Scene B, and **+100%** in Scene C. Qualitatively, V-GPS successfully resolved the failure modes discussed in Section 4. For example, on the "put pepper in pot" task in Scene A, which requires precise grasping, V-GPS **doubles** the success rate from 15% to 35%. Observe in Figure 4 that the robot can grasp the slippery pepper more reliably, leading to improved performance. Furthermore, V-GPS largely addresses the pre-mature and untimely release of objects in Scenes B and C. As an example, on the "put mushroom on cloth" task, incorporating the value function accurately up-

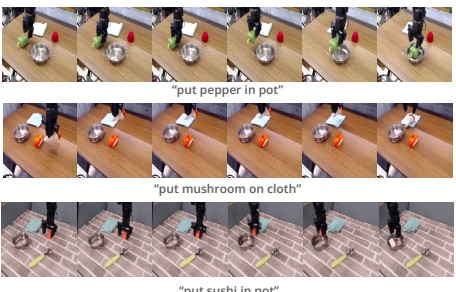

"put pepper in pot"

"put mushroom on cloth"

"put sushi in pot"

**Figure 4: (Qualitative visualizations)** V-GPS improves the precision of grasping the slippery object (first row), prevents the policy's default behavior of releasing the object too early (second row) and holding the object for too long (third row). More qualitative results and videos can be found at https://nakamotoo.github.io/V-GPS

weights the gripper close action until the mushroom is over the cloth, allowing the generalist policy to deviate from its default behavior of releasing the mushroom too early, as shown in Figure 4. This alone **doubles** the performance on this task. On the "put sushi in the pot" task in Scene C, Octo suffers from a late dropping issue, while V-GPS **triples** the performance on this task. This suggests that our value function can rank and select the critical action to drop the sushi at the right time.

## 6.3 Can V-GPS improve various generalist policies across different embodiments?

To answer this question, we now evaluate V-GPS on top of five generalist policies – Octo-small, Octo-base, Octo-small-1.5, RT1-X, and OpenVLA, in SIMPLER simulation environments. As shown in Table 2, our method improves all five policies across multiple embodiments on average. In

|  | Task | Octo-small-1.5 | V-GPS (Ours) | Improvement |
|---|---|---|---|---|
| Scene A | Green pepper in pot | 0.15 | **0.35** | |
| | Sweet potato on cloth | 0.30 | **0.35** | |
| | Average | 0.23 | **0.35** | +55.6% |
| Scene B | Mushroom on cloth | 0.35 | **0.70** | |
| | Mushroom in pot | 0.30 | **0.55** | |
| | Average | 0.33 | **0.63** | +92.3% |
| Scene C | Sushi in pot | 0.10 | **0.30** | |
| | Spoon in pot | 0.25 | **0.40** | |
| | Average | 0.18 | **0.35** | +100% |
| Total | Average | 0.24 | **0.44** | +82.8% |

**Table 1: (Real-world performance)** V-GPS consistently improves the success rates of Octo across the board, achieving an 82.8% improvement on average. This demonstrates that using our value function to re-rank the actions can enhance the generalist policy.

|  | Task | Octo-s | Octo-s +Ours | Octo-b | Octo-b +Ours | Octo-s-1.5 | Octo-s-1.5 +Ours | RT-1-X | RT-1-X +Ours | OpenVLA | OpenVLA +Ours |
|---|---|---|---|---|---|---|---|---|---|---|---|
| WidowX | Spoon on towel | 0.52 | 0.46 | 0.25 | 0.21 | 0.01 | 0.06 | 0.01 | 0.01 | 0.00 | 0.00 |
| | Carrot on plate | 0.15 | 0.16 | 0.18 | 0.24 | 0.00 | 0.00 | 0.06 | 0.07 | 0.06 | 0.04 |
| | Stack blocks | 0.07 | 0.07 | 0.00 | 0.01 | 0.00 | 0.02 | 0.00 | 0.00 | 0.00 | 0.02 |
| | Eggplant basket | 0.49 | 0.84 | 0.28 | 0.33 | 0.01 | 0.44 | 0.01 | 0.03 | 0.14 | 0.20 |
| | Average | 0.30 | **0.38** | 0.17 | **0.20** | 0.01 | **0.13** | 0.02 | **0.03** | 0.05 | **0.07** |
| Google Robot | Pick Can | 0.31 | 0.38 | 0.29 | 0.24 | 0.05 | 0.43 | 0.19 | 0.29 | 0.72 | 0.82 |
| | Put Near | 0.12 | 0.16 | 0.04 | 0.05 | 0.10 | 0.15 | 0.44 | 0.42 | 0.52 | 0.56 |
| | Average | 0.22 | **0.27** | **0.17** | 0.14 | 0.07 | **0.29** | 0.32 | **0.36** | 0.62 | **0.69** |
| Total | Average | 0.27 | **0.34** | 0.17 | **0.18** | 0.02 | **0.18** | 0.12 | **0.14** | 0.24 | **0.27** |

**Table 2: (SIMPLER [11] performance)** V-GPS improves the success rates of all five generalist policies across multiple embodiments using the same *single* value function.

particular, V-GPS improves the "put eggplant in basket" task by a large margin for all policies. As shown in Figure 3, unlike the other tasks that are open-space tabletop manipulation, this eggplant task is unique because it presents a vertical height difference and an obstructing wall between the target basket and the sink. As a result, any policy that is not careful would likely hit the wall and fail to complete the task. In addition, the slippery surface of the eggplant requires precise grasping locations and a carefully modulated grip to execute the task effectively. Therefore, the empirical finding that V-GPS produces the biggest benefits on this eggplant task aligns with our discussion in Section 4. In addition, Octo-small-1.5 performs surprisingly poorly in SIMPLER compared to its previous Octo-small model. Nonetheless, combining V-GPS with Octo-small-1.5 can largely mitigate its performance degradation. This might suggest that V-GPS effectively makes the policy robust against performance variations caused by changes in checkpoints or training recipes. In addition, We also show that V-GPS is preferred over fine-tuning the generalist policy itself in Appendix E.

# 7  Discussion and Future Work

In this paper, we presented V-GPS, an approach that utilizes value functions for steering generalist robot policies upon deployment. V-GPS does not require altering or fine-tuning the generalist policy, and can even operate effectively with black-box access to a pre-trained policy. Via thorough evaluation in both simulation and the real world, we show that V-GPS significantly improves the robustness and precision of pre-trained policies. Despite these promising results, there are still limitations. First, while V-GPS can, in principle, be combined with any policy, the policy must be able to sample actions stochastically rather than deterministically to generate diverse action candidates. Second, since V-GPS utilizes a separate value function, it does increase the computational and time expenses during deployment. While this is not a significant issue in our experiment (see Appendix G), it might limit its applicability in high-frequency tasks. Future work could explore achieving a "compute-optimal" balance between using the policy and querying the value function. Finally, while V-GPS can improve generalist policies on in-distribution tasks and environmental changes (e.g., table texture, height, etc.), its ability to handle completely unseen languages and objects is limited, as the value function is trained on data from only two robotic embodiments. Scaling up value function architectures and using more diverse data is a promising direction for future work.

**Acknowledgments**

We thank Zhiyuan Zhou for his help and suggestions on the implementation, and Seohong Park and Pranav Atreya for their informative discussions. We also thank Homer Walke, Karl Pertsch, and Xuanlin Li for providing details about Octo, OpenVLA, and SIMPLER. Additionally, we thank Noriaki Hirose for his feedback on the teaser figure.

This research was supported by the AI Institute, NSF FRR IIS-2150826, ONR N00014-20-1-2383, and AFOSR FA9550-22-1-0273. The computation was supported by the Google TPU Research Cloud (TRC) program through their TPU donations.

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

## A  V-GPS with IQL

While we used Cal-QL to train our value function in the main paper, one can use any offline RL or policy evaluation algorithm to fit $Q_\theta$. To demonstrate this, we further show the results with an IQL value function in this section. Formally, IQL trains a Q function $Q_\theta(s, a, l)$ and a state-value function $V_\psi(s, l)$ with the following objectives, where $L_2^\tau$ is the expectile loss $L_2^\tau(u) = |\tau - \mathbb{1}(u < 0)|u^2$, and $Q_{\bar\theta}$ represents the target Q-network, a delayed soft average of the current Q-network:

$$L_V(\psi) = \mathbb{E}_{(s,a,l)\sim\mathcal{D}}\left[L_2^\tau\left(Q_{\bar\theta}(s, a, l) - V_\psi(s, l)\right)\right] \tag{3}$$

$$L_Q(\theta) = \mathbb{E}_{(s,a,s',l)\sim\mathcal{D}}\left[\left(r(s, a, l) + \gamma V_\psi(s', l) - Q_\theta(s, a, l)\right)^2\right]. \tag{4}$$

We evaluated V-GPS using the IQL value function in SIMPLER. As shown in Table 3, using an IQL value function for V-GPS is also effective for improving the success rates of all five generalist policies across multiple embodiments.

| | Task | Octo-s | Octo-s +Ours | Octo-b | Octo-b +Ours | Octo-s-1.5 | Octo-s-1.5 +Ours | RT1-X | RT1-X +Ours | OpenVLA | OpenVLA +Ours |
|---|---|---|---|---|---|---|---|---|---|---|---|
| | Spoon on towel | 0.52 | 0.50 | 0.25 | 0.16 | 0.01 | 0.07 | 0.01 | 0.03 | 0.00 | 0.02 |
| | Carrot on plate | 0.15 | 0.18 | 0.18 | 0.20 | 0.00 | 0.00 | 0.06 | 0.07 | 0.06 | 0.06 |
| WidowX | Stack blocks | 0.07 | 0.09 | 0.00 | 0.00 | 0.00 | 0.02 | 0.00 | 0.00 | 0.00 | 0.00 |
| | Eggplant basket | 0.49 | 0.59 | 0.28 | 0.37 | 0.01 | 0.07 | 0.01 | 0.01 | 0.14 | 0.54 |
| | Average | 0.30 | **0.34** | 0.17 | **0.18** | 0.01 | **0.04** | 0.02 | **0.03** | 0.05 | **0.15** |
| Google Robot | Pick Can | 0.31 | 0.30 | 0.29 | 0.30 | 0.05 | 0.47 | 0.19 | 0.32 | 0.72 | 0.78 |
| | Put Near | 0.12 | 0.17 | 0.04 | 0.06 | 0.10 | 0.21 | 0.44 | 0.43 | 0.52 | 0.44 |
| | Average | 0.22 | **0.23** | 0.17 | **0.18** | 0.07 | **0.18** | 0.32 | **0.37** | **0.62** | 0.61 |
| Total | Average | 0.27 | **0.31** | 0.17 | **0.18** | 0.02 | **0.14** | 0.12 | **0.15** | 0.24 | **0.31** |

**Table 3: (V-GPS with IQL)** Using an IQL value function for V-GPS is also effective for improving the success rates of all five generalist policies across multiple embodiments.

## B  V-GPS Implementation Details

In this section, we provide the implementation details of V-GPS for value function pre-training, and test-time action re-ranking. The hyperparameters are listed in Table 4.

### B.1  Value Function Training

Our language-conditioned Q function $Q_\theta(s, a, l)$ uses a ResNet-34 image encoder with FiLM language conditioning as shown in Figure 5. The image observation is first passed through the ResNet-34 encoder, while the language instruction, processed by a frozen MUSE encoder, is applied to every block in ResNet using FiLM conditioning. The 7-dimensional actions are concatenated with the final output from the ResNet, then passed through two 256-unit hidden layers, and finally, a scalar Q value is predicted. For both Cal-QL and IQL, we trained the value function using a mixture of the Bridge and Fractal datasets with a batch size of 512 on a single v4-8 TPU VM. We used a discount factor $\gamma = 0.98$, clipped double Q-learning [64], and shifted reward values of 0 and $-1$. We assigned the final 3 steps of each trajectory as positive rewards 0, and the rest as negative rewards $-1$. We use the Adam optimizer with a learning rate of 3e-4. During training, we augment the image observations with random cropping and color jitter. The Cal-QL value function is trained using an alpha of $\alpha = 5.0$ for 1M steps. The IQL value function is trained using an expectile of $\tau = 0.7$ for 200K steps.

### B.2  Test-Time Action Re-Ranking

During test-time, we sample $K$ action proposals from the base policy $\pi$ at each time step, and then re-rank the proposed actions using the Q function with Equation 2. In the real-world evaluations with the Cal-QL value function, we used $K = 50$ and we found selecting the action greedily by setting $\beta \to 0$ leads to satisfactory results. In simulation, we swept over $K = \{10, 50\}$ and $\beta = \{0, 0.1, 1.0\}$ and report the best result for each policy.

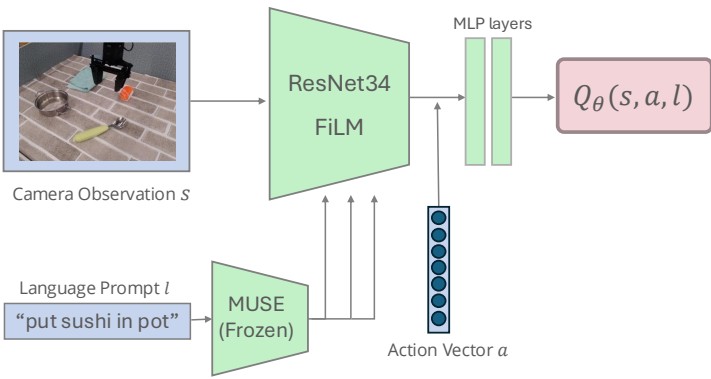

**Figure 5: (Model Architecture.)** Our value function uses a ResNet-34 image encoder with FiLM language conditioning.

| | |
|---|---|
| Cal-QL $\alpha$ | 5.0 |
| IQL expectile $\tau$ | 0.7 |
| discount factor | 0.98 |
| learning rate | 3e-4 |
| positive reward steps $H$ | 3 |
| number of actions to sample $K$ | $\{10, 50\}$ |
| softmax temperature $\beta$ | $\{0, 0.1, 1.0\}$ |

**Table 4: (V-GPS hyperparameters)**

## C  Baseline Implementation Details

For Octo-{small, base, small-1.5}, we used the publicly released checkpoints from `https://huggingface.co/rail-berkeley`. For RT1-X, we used the publicly released JAX checkpoint from `https://github.com/google-deepmind/open_x_embodiment`. For OpenVLA, we used their public checkpoint `openvla-v01-7b` from `https://huggingface.co/openvla/openvla-v01-7b`. To combine OpenVLA with our method, we had to iterate the forward pass $K$ times at each time step to sample multiple actions, since it does not yet support batch inference. Our real-world evaluation is implemented on top of the evaluation codes provided from `https://github.com/octo-models/octo`, and the simulated evaluation is based on `https://github.com/simpler-env/SimplerEnv`.

## D  Experimental Setup

**(Real world)** We conducted our real-world evaluations on 6 tasks across 3 different scenes as shown in Figure 3. We provide the language instructions we used for each task in Table 5. We conduct 20 trials per task and report the average success rates in Table 1. We randomize the configurations and orientations of each object for each trial.

**(SIMPLER)** We conducted the simulated evaluations on 6 tasks in the SIMPLER environment, including 4 tasks on the WidowX robot platform and 2 on the Google Robot platform as shown in Figure 3. We used the default language instructions for each task as shown in Table 6. For RT1-X and Octo-{small, base, small-1.5}, we conducted 100 trials for each of three different random seeds. For OpenVLA, we conducted 50 trials per task due to its slower inference speed, as it does not yet support batch inference. The average success rates are reported in Table 2.

| | Language Instructions |
|---|---|
| Scene A | put the green pepper in the pot 
 put the sweet potato on the cloth |
| Scene B | put the mushroom on the cloth 
 put the mushroom in the pot |
| Scene C | put the sushi in the pot 
 put the green spoon in the pot |

**Table 5: (Real-world scenes and tasks)** We evaluate V-GPS in 6 tasks across 3 different real-world scenes.

| | Language Instructions |
|---|---|
| WidowX | put the spoon on the towel 
 put carrot on plate 
 stack the green block on the yellow block 
 put eggplant into yellow basket |
| Google Robot | pick coke can 
 move {object1} near {object2} |

**Table 6: (SIMPLER scenes and tasks)** We evaluate V-GPS in 6 tasks across 2 different embodiments in SIMPLER environment.

**(Further Details)** Our goal is to use a value function to improve the pre-trained generalist policies, and we do not assume access to any additional data beyond what the generalist policies were pre-trained on. All the generalist policies are pre-trained on the OXE dataset, which is fully open-sourced and public. Our experiments are designed to study the following cases:

- SIMPLER: clearly in-distribution, with the same combination of objects in the same scene
- Scene A: seen tabletop, with different combinations of objects
- Scene B: seen tabletop with a lower table height of 1 inch than usual (to test distribution shift), and with different combinations of objects
- Scene C: unseen tabletop, with different combinations of objects

Given that V-GPS improves upon the pre-trained policy in SIMPLER and in Scenes A, B, and C, our method has proven to be effective in both in-distribution cases and domain shifts, including changes in table height and unseen tabletops or backgrounds.

## E  Additional Comparisons to Policy Fine-Tuning Approaches

A common question is: Why is re-ranking with Q-values preferred over fine-tuning generalist policies? There are several reasons why re-ranking with Q-values might be more effective. First, large generalist models might be closed-source and available via API only (such as RT-2-X), which hinders fine-tuning. Second, as these models increase in size, fine-tuning becomes increasingly computationally expensive. Fine-tuning OpenVLA, for instance, requires $8 \times$ A100s.

Furthermore, since the generalist policies are pre-trained on the OXE dataset, which already contains the datasets for the downstream tasks we are studying (the Bridge dataset and the Fractal dataset), further fine-tuning the generalist policy on these individual datasets does not necessarily improve performance. To demonstrate this, we conducted additional experiments to compare V-GPS to three different policies:

1. **Octo-finetune**: Octo-small model pre-trained on OXE + fine-tuned on bridge dataset
2. **Octo-scratch**: Octo-small model trained on bridge dataset from scratch
3. **Resnet-DP**: Diffusion Policy [37] with a Resnet34 encoder, which is a state-of-the-art architecture for imitation learning, trained on bridge dataset from scratch

As shown in Table 7, neither fine-tuning the generalist policy nor training a policy solely on a single dataset improves performance over Octo, and V-GPS is the only method that achieves better performance than the generalist policy. This clearly highlights the benefit of re-ranking with Q-values preferred over fine-tuning the generalist policies.

## F  Ablation Over the Size of Dataset

To investigate how the size of the offline dataset impacts performance, we trained IQLvalue functions on smaller datasets – Bridge Dataset subsampled to 50% and 10% – and evaluated the performance on SIMPLER's eggplant task. As shown in Table 8, reducing the dataset size to 50% still

| Task | Octo-small | Octo-finetuned | Octo-scratch | Resnet-DP | Ours (IQL) | Ours (Cal-QL) |
|---|---|---|---|---|---|---|
| Spoon on towel | 0.52 | 0.28 | 0.01 | 0.05 | 0.50 | 0.46 |
| Carrot on Plate | 0.15 | 0.12 | 0.01 | 0.01 | 0.18 | 0.15 |
| Stack blocks | 0.07 | 0.06 | 0.00 | 0.06 | 0.09 | 0.07 |
| Eggplant basket | 0.49 | 0.41 | 0.00 | 0.37 | 0.59 | 0.84 |
| Average | 0.30 | 0.22 | 0.01 | 0.12 | **0.34** | **0.38** |

Table 7: (**Comparison to fine-tuning generalist policies or training the policy from scratch.**) V-GPS is the only method that achieves better performance than the generalist policy.

achieved the same improvement, and reducing it to 10% resulted in slightly worse performance, but it still improved over Octo-small. This shows that even a value function trained on small amounts of data can be effective in guiding generalist policies at test time.

| Model | Success Rate |
|---|---|
| Octo-small (baseline) | 0.49 |
| Ours-100% | 0.59 |
| Ours-50% | 0.59 |
| Ours-10% | 0.55 |

Table 8: (**Ablation over the size of datasets.**) Even a value function trained on small amounts of data can be effective in guiding generalist policies at test time.

# G   Analysis of the Overhead in Inference Time

We conducted an analysis of the inference time per time step using the Octo-small model. As shown in Table 9, using $K = 10$ results in 1.28 times slower inference, and using $K = 50$ results in 1.59 times slower inference compared to the baseline, which we did not find to be a significant slowdown in practice. The analysis is conducted on the inference machine that we used for real-world evaluation. Furthermore, this level of overhead will not be an issue in real-world WidowX tasks. This is because the WidowX environment typically uses blocking control with a 0.2-second interval, meaning actions are predicted every 0.2 seconds [19, 9].

| Method | Inference time (s) | Overhead |
|---|---|---|
| Octo-small | 0.0752 | 1.00 |
| Ours $K = 10$ | 0.0963 | 1.28 |
| Ours $K = 30$ | 0.1096 | 1.46 |
| Ours $K = 50$ | 0.1196 | 1.59 |
| Ours $K = 100$ | 0.1596 | 2.12 |

Table 9: (**Analysis of the overhead in inference time.**)

# H   Ablation Over the Number of Actions $K$

We conducted an ablation study on $K$ using the WidowX eggplant task and the Google Robot pick-coke task. As shown in Table 10, we found that the IQL value function performs best with $K = 10$, and increasing $K$ leads to the exploitation of the value function, resulting in performance degradation. In contrast, the Cal-QL value function is more robust to $K$, and using a larger value can improve performance.

| Task | Eggplant | | Pick Coke | |
|---|---|---|---|---|
| Offline RL method | IQL | Cal-QL | IQL | Cal-QL |
| Octo-small (baseline) | 0.49 | 0.49 | 0.31 | 0.31 |
| Ours $K = 10$ | 0.59 | 0.77 | 0.30 | 0.38 |
| Ours $K = 30$ | 0.47 | 0.81 | 0.37 | 0.38 |
| Ours $K = 50$ | 0.42 | 0.84 | 0.31 | 0.38 |
| Ours $K = 100$ | 0.35 | 0.63 | 0.37 | 0.36 |

Table 10: (**Ablation over $K$.**) Cal-QL is more robust to $K$ than IQL.

# I Comparisons to the Actors of Cal-QL & IQL

We evaluated the IQL and Cal-QL actors in the SIMPLER tasks, but as shown in Table 11, we found that they were unable to successfully complete them, consistently achieving a zero success rate. Interestingly, the common failure case for these actors was their inability to learn the gripper's proper opening and closing actions, consistently outputting the open-gripper action. We also tried to roll out the actors in the real-world setup and observed the same issue. This is a common problem when training a Gaussian (or Tanh-squashed Gaussian) actor on manipulation datasets such as Bridge Data, since the modes of the gripper's open/close distribution are too extreme (0 for close and 1 for open), and the actor is too simple to model them effectively. This highlights the benefit of our method, which combines the value function with the pre-trained generalist policies, allowing us to enjoy the advantages of both the critic and the state-of-the-art expressive imitation learning policies.

| Task | IQL actor | Cal-QL actor |
|---|---|---|
| Spoon on towel | 0.00 | 0.00 |
| Eggplant basket | 0.00 | 0.00 |

Table 11: (**Comparisons to the actors of Cal-QL & IQL.**) The actors of Cal-QL and IQL consistently achieve a zero success rate. This highlights the benefit of our method, which combines the value function (critic) with the pre-trained generalist policies.

# J Comparison to Using a Random Policy or Random Action Selection

To prove that both parts of V-GPS – the value function and the generalist policy – are the specific reasons for improvement, we compared the following two methods on the eggplant task:

1. **Random-selecting**: Octo-small policy + randomly selecting actions
2. **Random-policy**: Random policy + V-GPS value function

The results are shown in Table 12. As expected, Random-selecting performs similarly to the naive Octo-small model, showing no improvement. This highlights the benefit of using the value function for action selection. Furthermore, Random-policy fails to perform the task and consistently achieves a zero success rate. This is also expected, as if the policy generates nonsensical action proposals, then using the value function will not help. In short, both parts of V-GPS – the pre-trained policy and the value function – are crucial for improvement, and combining them both together leads to the best performance.

| Method | Success Rate |
|---|---|
| Octo-small (baseline) | 0.49 |
| Random-selecting | 0.49 |
| Random-policy | 0.00 |
| V-GPS (ours) | 0.84 |

Table 12: (**Comparison to using a random policy or selecting the actions randomly.**) Using a random policy or random action selection does not improve performance over the generalist policy, demonstrating that V-GPS is the specific reason for the improvement.

# K Details of the Network Size

We provide the number of parameters for our value function and the generalist policies in Table 13. Our value function is a ResNet-34-based network with 25.6 million parameters. This is smaller than all the generalist policies we studied, specifically 27% the size of the Octo-base model and 0.3% the size of Open-VLA.

| Model | Num Params |
|---|---|
| Q Network (Ours) | 25.6M |
| Octo-small | 27M |
| Octo-base | 93M |
| OpenVLA | 7B |
| RT1-X | 35M |

Table 13: (**Network size.**) Our critic network is smaller than all the generalist policies we studied, specifically 27% the size of the Octo-base model and 0.3% the size of OpenVLA.

