# OpenReview forum: "Steering Your Generalists: Improving Robotic Foundation Models via Value Guidance"
_robot-learning.org/CoRL/2024/Conference — CoRL 2024_

### Official Review · Reviewer_jKak · 2024-07-21
**Review of Submission 669**

**Originality:** 4
**Technical Quality:** 3
**Clarity Of Presentation:** 4
**Potential Impact:** 3
**Recommendation:** 2
**Confidence:** 4

**Review:**

The primary insight of the paper is that re-ranking multiple action proposals from a generalist policy using a trained value function as test time improves performance. Overall, I find their approach interesting and relatively novel, but the evaluations are not impressive and do not provide insights on what helps improve policy performance.

Strengths
1. The authors test their system in the real world and across multiple embodiments in simulation.
2. V-GPS is an intuitive approach that could be tacked onto any generalist policy, which makes it a compelling method
3. V-GPS is shown to have large performance improvements in the real world and in simulation.

Weaknesses
1. As the authors note, their approach uses a large dataset (45k samples from Bridge v2). Would reducing the dataset size impact V-GPS performance?
2. The motivation of this work is to improve the policy performance of generalist policies, and V-GPS is meant to do so by re-ranking multiple action proposals. However, the authors do not compare their method with other possible techniques. Comparisons with methods such as a policy trained solely on Bridge v2, the full policy fine-tuned, or a parameter-efficient finetuning of the policy would be insightful. If the goal is not to show other approaches to improving performance, it would also be useful to have other methods that provide scores for different action proposals, either based on heuristics or simple random sampling would provide insights on what leads to the increase in performance. The results do not allow one to *conclude* that V-GPS is the specific reason why the performance improved—it might be due to the additional data, the fact the data was trained on more recently, the action selection process, etc.
3. One hypothesis is that the generalist policy maintains semantic generalization knowledge while V-GPS can focus on ensuring the lower-level actions perform well. How much work is V-GPS doing for improving policy performance over the generalist policy alone? If you replaced the generalist policy with a random policy and V-GPS, would the performance be similar?
4. The need for multiple forward passes of the generalist model increases computational demands and affects real-time capabilities. Are the K sampled actions processed sequentially or in a batch? How much overhead does V-GPS introduce?
5. The impact of the language-conditioned aspect on performance is not thoroughly explored.

Typos:
Line 56: "Vlue-Guided" -> "Value-Guided"

**Quality Of The Limitations Section:**

3

**Questions For Rebuttal:**

- The authors' real world experiments were on the Bridge data. Was Bridge included in the training data for the generalist policies used? Does the SIMPLER simulation replicate environments that are found in the training data used in the generalist policies?
- What is the relationship of this kind of work to methods like LOReL's visual model predictive control [1] which samples M different actions, predicts a reward for each action sequence with a language-conditioned visual reward function, then executes the best one which they find by using the Cross Entropy Method (CEM). In your case, it appears that a large generalist model provides a few initial actions proposals instead, then a similar approach to LOReL ranks the action proposals.
- The authors use the term "generalist value functions". In the real-world experiments, only Bridge data is used, and in the SIMPLER simulation experiments a mix of Bridge and Fractal data is used. Are two embodiments sufficient to be a generalist value function? I did not put this question under weaknesses as I do not think the robotics community has converged to what it means to truly be a "generalist" policy, so I am curious what the authors think.
- More questions are in the weaknesses section above.

[1] S. Nair, E. Mitchell, K. Chen, S. Savarese, C. Finn, et al. Learning language-conditioned robot behavior from offline data and crowd-sourced annotation. CoRL 2021.

**Robotics Focus:**

4

**Summary Of Paper:**

Generalist robot policies perform sometimes have issues with closing the gripper at the right time, releasing the gripper on time, etc. The authors train a value function to re-rank multiple sampled actions and execute the action that the value function thinks is likely to succeed. They do this by training a language-conditioned value function that they train. This value function is agnostic to the weights of the full policy and can be applied to any robot policy. It does so by re-ranking actions and choosing the best one.

**Summary Of Recommendation:**

Although the paper has good real-world deployments, I would recommend this paper in its current state for a weak reject due to the lack of comparisons and limited discussion of any other kinds of methods. The idea of test-time action re-ranking is interesting. However, by not evaluating any potential alternatives, this work does not sufficiently show that it is better than other approaches. I am open to discussion on these points and increasing my score.

---

### Official Review · Reviewer_AHDo · 2024-07-21
**A novel approach to improving robotic foundation models without having to retrain them. The paper is well-structured and well motivated and the results are comprehensive. A few ablations will help.**

**Originality:** 3
**Technical Quality:** 3
**Clarity Of Presentation:** 3
**Potential Impact:** 3
**Recommendation:** 3
**Confidence:** 4

**Review:**

Strength:
The paper is well written, well motivated and the idea is fairly novel. It is desirable to preserve the generalist policy and simply steer it.

Weakness Major:
The paper is similar to Say-Can, a seminal paper that uses Value functions to reweigh the probabilities of actions from an LLM. I understand V-GPS is for raw sampled actions in joint space as opposed to Say-Can’s actions in skill space, but the similarity suggests that it be included in the related works sections, with a brief description of how V-GPS is different.

An ablation over K would be interesting. It’s possible that sampling more actions can lead to better solutions after reweighting, but this might be a tradeoff for execution time.

IQL not only trains a Q function but also a policy to go with it. What was the performance of the policy that was trained with IQL? Can you provide an analysis comparing V-GPS and the IQL policy? The fact that the Q function works so well in ranking actions suggests that the state distribution of the IQL policy might match that of the foundation model.


Minor: confusing notation K used for both Q function training and action sampling.

**Quality Of The Limitations Section:**

3

**Questions For Rebuttal:**

What was the size of the Critic Q network used? Does scaling this model lead to better results? If the Q network is much smaller than the pre-trained foundation model, then this enables researchers with lesser compute to train these models instead of fine-tuning much larger foundational models. This fact can strengthen the paper if explicitly mentioned.

What exactly was the overhead added to the inference time in VGPA compared to the regular Octo/VLA models? This is mentioned as a limitation, but knowing the exact number might be useful. Please club this with the ablation over K mentioned above.

**Robotics Focus:**

4

**Summary Of Paper:**

The paper presents a fairly unique approach (V-GPS) to improving pre-trained foundation models for manipulation. The method first trains a value function with IQL. V-GPS then ranks sampled actions from a pre-trained foundation model by the estimated q value and then employs those actions. The paper shows that this procedure improves the performance of the foundation model, without having to pass policy gradients through it.

**Summary Of Recommendation:**

The paper is novel and well written, providing ablations will strengthen it further.

---

### Official Review · Reviewer_Gdoc · 2024-07-26

**Originality:** 4
**Technical Quality:** 3
**Clarity Of Presentation:** 4
**Potential Impact:** 4
**Recommendation:** 2
**Confidence:** 4

**Review:**

- Strengths
    - The proposed method can be combined with different pretrained policies to improve their performance.
    - The improvements have been demonstrated in simulated robot and real robot settings.
    - The analysis of the improvements of failure modes shows the value functions can help action selection.

- Weaknesses
    - While the using learned value function to improve action selection is a compelling idea, I found it is unclear when the re-ranking of the actions is useful is still unclear. If it can improve the in-distribution cases, the reported numbers on the pretrained Octo and OpenVLA are much worse than the numbers reported in their original report which makes it harder to interpret the results. If it mainly improves the distribution shift cases, the current experiments seem to be very close to the Open-X-Embodiment (OXE) datasets the generalist models are trained on.
    - To obtain the Q-values, we need to get an offline dataset to learn the value functions. While the paper uses a mix of Bridge Data and Google robot datasets to train the value function, in the real environment, it may be hard to get a dataset as large as those datasets to learn the value function. In such cases, it is hard to understand the impact of the offline dataset on policy improvements. If we can already get a large offline dataset, why re-ranking with Q-values is preferred over fine-tuning the generalist policies?

**Quality Of The Limitations Section:**

3

**Questions For Rebuttal:**

- To better support the claim of policy improvements, it will be helpful if the authors can split the experiment into different cases such as in-distribution, distribution shifts, and different embodiments to answer when and how the proposed method can help the generalist policies.
- The assumption about a good offline dataset may not be realistic, it will be helpful to see an experiment on how the size of the offline dataset could impact the performance improvements.
- The current experiments show good improvements in WindowX settings. However, it doesn’t seem to help Google robot that much, especially negative impact in the OpenVLA case. Why is there a gap between different embodiments?

**Robotics Focus:**

4

**Summary Of Paper:**

This paper present, V-GPS, a value-based reranking method to improve the quality of the pretrained generalist policy. This approach is to fix the policy failures due to imprecise manipulation at the testing time. The proposed approach leverages offline RL to learn the Q-value of the test domain. Then using the Q-value to rerank the top K actions. The experiments showed big improvements in simulated and real robot tasks.

**Summary Of Recommendation:**

The overall idea is interesting and can have good impact for the generalist policy deployment. As discussed in weaknesses and questions, some questions still need to be answered to support the claim.

---

### Author Rebuttal · Authors · 2024-08-12

## Summary of Contributions
---

### We thank all the reviewers and the area chair for their thoughtful comments and are pleased that our work was received positively! Below, we summarize our newly added contributions with a PDF file showing the additional results. Each table can also be found in the individual responses within the official comments.
---
**Additional Evaluation Experiments**

As requested by the reviewers, we have added new experiments showcasing that
1. V-GPS performs better than fine-tuning the generalist policies in Table 1
1. Dataset size ablation showing that even a value function trained on small amounts of data can be effective in guiding generalist policies at test time in Table 2
1. Analysis of the overhead in inference time in Table 3
1. An ablation over K samples that leads to the observation that conservative-based methods are more robust to K in Table 4
1. Evaluation of IQL actor in Table 5
1. Comparisons of to random scoring of action + generalist policies and random policy + V-GPS in Table 6
---

**Additional Discussions for Clarity**

In each of the individual responses, we have provided responses to the reviewer's questions and concerns to improve clarity, expanding our presentation of:
- When does the re-ranking of actions help?
- Details of evaluation tasks to clarify how in/out-distribution they are.
- Details of the training data and the clarification that our method does not assume any additional offline data
- Details of the network size of our critic Q network and the generalist policies
- Clarification regarding the mismatch with the original papers
---
We will add these additional experimental results and clarifications to the final version of the paper. We believe that we have addressed all the concerns raised by the reviewers, but if any further concerns remain, please let us know!

---

### Decision · Program_Chairs · 2024-09-04

**Decision:**

Accept

**Comment:**

Strengths
- Experiments show significant improvements over baselines
- Can be combined with different pre-trained policies
- Innovative idea

Weaknesses
- Unclear in which cases the method is helpful
- Reported results don't match with original papers
- Large offline dataset might be unrealistic
- Some missing details
- Experiments insufficient to prove that V-GPS was the specific reason for improvement

# After discussion phase:

The authors successfully managed to address the most critical concerns of the reviewers. They are now all voting "weak accept" (but some couldn't update their original scores). All reviewers find the method interesting and novel. The improvement over the SOTA is not massive (also see below), but the idea behind the paper definitely deserves to be published as opens avenues for future research without requiring massive compute.

During the reviewer and AC discussion phase one more issue popped up that needs to be fixed: Reporting relative improvement is probably not the best option for this paper. This seems to be totally skewed by octo-s-1.5, where performance increases from 1% to 7% leads to 700% relative improvements (marginally improving a bad generalist policy) that brings the overall relative improvement to a high number, while absolute improvement is a lot more modest.